# Bilevel Learning of the Group Lasso Structure

**Jordan Frecon**[*,1]     **Saverio Salzo**[*,1]     **Massimiliano Pontil**[1,2]
[1] Computational Statistics and Machine Learning, Istituto Italiano di Tecnologia (Italy)
[2] Department of Computer Science, University College London (UK)

## Abstract

Regression with group-sparsity penalty plays a central role in high-dimensional prediction problems. However, most existing methods require the group structure to be known *a priori*. In practice, this may be a too strong assumption, potentially hampering the effectiveness of the regularization method. To circumvent this issue, we present a method to estimate the group structure by means of a continuous bilevel optimization problem where the data is split into training and validation sets. Our approach relies on an approximation scheme where the lower level problem is replaced by a smooth dual forward-backward algorithm with Bregman distances. We provide guarantees regarding the convergence of the approximate procedure to the exact problem and demonstrate the well behaviour of the proposed method on synthetic experiments. Finally, a preliminary application to genes expression data is tackled with the purpose of unveiling functional groups.

## 1    Introduction

With recent technological advances, high-dimensional datasets have become massively widespread in numerous applications ranging from social sciences to computational biology [20, 25, 1]. In addition, in many statistical problems, the number of unknown parameters can be significantly larger than the number of data samples, thus leading to underdetermined and computationally intractable problems. Nonetheless, many classes of datasets exhibit a sparse representation when expressed as a linear combination of suitable dictionary elements. This has led, over the past decades, to the development of sparsity inducing norms and regularizers to unveil structure in the data. However, beyond the sparsity patterns of the data, there might also be a more complex structure, which is widely referred to as *structured sparsity* [16, 22, 23, 29]. In this line of research, a lot of work has been devoted to encode *a priori* structure of the data in (possibly overlapping) groups or hierarchical trees [31, 15, 17].

In the present paper, we restrict our study to the popular Group Lasso problem [30]. Given a vector of outputs $y \in \mathbb{R}^N$ and a design matrix $X \in \mathbb{R}^{N \times P}$, the Group Lasso problem amounts in finding

$$\hat{w} \in \underset{w \in \mathbb{R}^P}{\operatorname{argmin}} \frac{1}{2} \|y - Xw\|^2 + \lambda \sum_{l=1}^{L} \|w_{\mathcal{G}_l}\|_2, \tag{1}$$

for some regularization parameter $\lambda > 0$ and a non-overlapping group structure, i.e., an unordered partition of the features in $L$ groups $\{\mathcal{G}_1, \ldots, \mathcal{G}_L\}$ such that $\cup_{l=1}^{L} \mathcal{G}_l = \{1, \ldots, P\}$ and $(\forall l \neq l'), \mathcal{G}_l \cap \mathcal{G}_{l'} = \emptyset$. The specific form of the regularizer permits to enforce sparsity at the group-level, thus often leading to a better interpretability of the features than standard Lasso.

However, in many applications, we might have hundreds or thousands of features whose group-structure $\{\mathcal{G}_1, \ldots, \mathcal{G}_L\}$ may be unknown, or only partially known. In addition, the number of groups $L$ itself might not be known. Nonetheless, the prior knowledge of the group structure is crucial in order to achieve a lower prediction error [19]. Note that this problem can be seen as purely

---

[*]Equal contribution.

combinatorial since it amounts in searching the best partition amongst $(L^P/L!)$ possible unordered partitions.

In this paper we address the problem of learning the Group Lasso structure, within the setting of multi-task learning, through a bilevel optimization approach. We establish some basic mathematical properties of this methodology and demonstrate that it works well in practice.

**Related works.** We are only aware of few approaches devoted to infer the Group Lasso structure. A probabilistic modeling approach has been investigated in [14] to learn the relevance of pairs of features only. More recently, [27] considered a broad family of heavy-tailed priors for the group variables along with a variational inference scheme to learn the parameters of these priors. However, this approach becomes prohibitive when dealing with a large number of features. In addition, the setting is different: it analyzes the *latent* Group Lasso and it assumes that the learning tasks have a similar structure, meaning that the relevance of a given group is largely shared across the tasks.

**Contributions and outline.** The principal contribution of this paper is the formulation of the problem of learning the Group Lasso structure as a continuous bilevel optimization problem. In Section 2, we present, in a formal way, our bilevel approach. A new algorithmic solution based on an upper stochastic gradient descent and a lower dual forward-backward scheme with Bregman distances is devised in Section 3. The performance of the proposed approach are quantitatively assessed on synthetic data in Section 4, and shown to favorably compare against standard approaches. In addition, an application to real data in the context of gene expression analysis is provided with the goal of discovering functional groups. Finally, conclusions and perspectives are drawn in Section 5.

**Notations.** Let $\mathcal{X}$ be an Euclidian space. $\Gamma_0(\mathcal{X})$ denotes the space of functions $h\colon \mathcal{X} \to\, ]-\infty, +\infty]$ closed, proper and convex. We also denote by $\operatorname{argmin} h$ the set of minimizers of $h$ or the minimizer of $h$ when it is unique.

## 2 Proposed Bilevel Problem for Learning the Groups

In this section, we describe a bilevel framework for estimating the Group Lasso structure based on a multi-task learning problem, without any further *a priori* information.

### 2.1 Original Problem

We encapsulate the group structure by means of an hyperparameter $\theta = [\theta_1 \ldots \theta_L] \in \{0, 1\}^{P \times L}$, defining at most $L$ groups, such that $(\forall p \in \{1, \ldots, P\}, \forall l \in \{1, \ldots, L\})$, $\theta_{p,l} = 1$ if the $p$-th feature belongs to the $l$-th group, and 0 otherwise. Note that when no prior information on the number of groups is given, one should consider the extreme setting where there might be at most $L = P$ groups. In order to properly select $\theta$, we propose to consider the following bilevel problem.

**Problem 2.1** (Mixed Integer-Continuous Bilevel Problem.). *Given some vectors of outputs $y_t \in \mathbb{R}^N$ and design matrices $X_t \in \mathbb{R}^{N \times P}$ for $t \in \{1, \ldots, T\}$, as well as a regularization parameter $\lambda > 0$, find*

$$\hat{\theta} \in \operatorname*{argmin}_{\theta \in \{0,1\}^{P \times L}} C(\hat{\mathbf{w}}(\theta)) \quad s.t. \quad \sum_{l=1}^{L} \theta_l = \mathbb{1}_P, \tag{2}$$

*where $C(\hat{\mathbf{w}}(\theta)) = (1/T) \sum_{t=1}^{T} C_t(\hat{w}_t(\theta))$, $C_t \colon \mathbb{R}^P \to \mathbb{R}$ is a smooth function and $\hat{\mathbf{w}}(\theta) = (\hat{w}_1(\theta), \ldots, \hat{w}_T(\theta))$ is a minimizer of $T$ separated Group Lasso problems sharing a common group structure, i.e., it solves*

$$\operatorname*{minimize}_{(w_1,\ldots,w_T) \in \mathbb{R}^{P \times T}} \frac{1}{T} \sum_{t=1}^{T} \left( \frac{1}{2} \|y_t - X_t w_t\|^2 + \lambda \sum_{l=1}^{L} \|\theta_l \odot w_t\|_2 \right), \tag{3}$$

*where $\odot$ denotes the Hadamard product.*

The constraint in the right-hand side of Problem (2) ensures that every feature belongs to a single group. A natural choice for $C_t$ is to consider the validation error $C_t(\hat{w}_t(\theta)) = \frac{1}{2}\|y_t^{(\mathrm{val})} - X_t^{(\mathrm{val})} \hat{w}_t(\theta)\|^2$ evaluated on a set $\{y^{(\mathrm{val})}, X^{(\mathrm{val})}\}$. For such choice, the selection of $\theta$ is motivated by the need of generalizing well to unseen data. In practice, it is often a good surrogate to the estimation error when the true features are unknown.

We note that directly solving Problem 2.1 is a challenge, since it is a mixed integer-continuous bilevel problem. To overcome this difficulty, we consider in the next section a relaxation of the problem in the continuous setting.

## 2.2 Relaxed Problem

We propose to consider a continuous relaxed version of Problem 2.1 where $\theta \in [0,1]^{P \times L}$ and the penalty term $\frac{\epsilon}{2T} \sum_{t=1}^{T} \|w_t\|_2^2$ is added to (3), for some $\epsilon > 0$, in order to ensure strong convexity of the lower level objective function and hence the uniqueness of its minimizer. The resulting problem lies within the framework of continuous bilevel optimization [10] which has recently been gaining a renewed interest in image processing [18, 5] as well as in neural networks and machine learning (see e.g. [4, 21, 24, 12]). Our relaxation of Problem 2.1 is formulated as follows.

**Problem 2.2** (Exact Bilevel Problem). *Let $C$ be as in Problem 2.1 and let $\psi, \xi : [0,1] \to \mathbb{R}$ be increasing continuous functions such that $\psi(0) = \xi(0) = 0$ and $\psi(1) = \xi(1) = 1$. Given some vectors of outputs $y_t \in \mathbb{R}^N$ and design matrices $X_t \in \mathbb{R}^{N \times P}$ for $t \in \{1, \ldots, T\}$, as well as some regularization parameters $\lambda > 0$ and $\epsilon > 0$, solve*

$$\underset{\theta \in \Theta}{\text{minimize }} \mathcal{U}(\theta) \quad \text{with} \quad \begin{cases} \hat{w}(\theta) = \text{argmin}_{w \in \mathbb{R}^{P \times T}} \ \mathcal{L}(w, \theta), \\ \mathcal{U}(\theta) = C(\hat{w}(\theta)), \end{cases} \tag{4}$$

*where*

$$\begin{cases} \mathcal{L}(w, \theta) := \dfrac{1}{T} \sum_{t=1}^{T} \mathcal{L}_t(w_t, \theta), \quad \mathcal{L}_t(w_t, \theta) = \dfrac{1}{2} \|y_t - X_t w_t\|_2^2 + \dfrac{\epsilon}{2} \|w_t\|_2^2 + \lambda \sum_{l=1}^{L} \|\psi(\theta_l) \odot w_t\|_2, \\ \Theta = \left\{ \theta = [\theta_1 \ldots \theta_L] \in [0,1]^{P \times L} \ \Big| \ \sum_{l=1}^{L} \xi(\theta_l) = \mathbb{1}_P \right\}. \end{cases} \tag{5}$$

*and $\psi$ and $\xi$ are applied component-wise to the vectors $\theta_l$'s.*

**Remark 2.1.** The functions $\phi$ and $\xi$ permit to cover different continuous relaxations of Problem 2.1 and the conditions $\psi(0) = \xi(0) = 0$ and $\psi(1) = \xi(1) = 1$ are compatibility conditions with Problem 2.1. Among the different choices of $\psi$ and $\xi$ we point out $\psi = \xi = \text{Id}$, which corresponds to a convex relaxation in which $\Theta$ is the unit simplex.

The following result establishes the existence of solution of Problem 2.2. The proof is given in the supplementary material (Section A.1).

**Proposition 2.1** (Existence of Solutions). *Suppose that $\Theta$ is a compact nonempty subset of $\mathbb{R}_+^{P \times L}$ and that $C$ and $\psi$ are continuous functions. Then $\theta \mapsto \hat{w}(\theta)$ is continuous and hence Problem 2.2 admits solutions.*

## 2.3 Approximate Problem

Usually, we don't have a closed form expression for $\hat{w}(\theta)$ but we rather have an iterative procedure converging to $\hat{w}(\theta)$ that we arbitrarily stop after $Q$ iterations. Therefore, we actually solve an approximate problem of the following form.

**Problem 2.3** (Approximate Bilevel Problem). *Let $C$ and $\Theta$ be as in Problem 2.2. Given two mappings $\mathcal{A}$ and $\mathcal{B}$, as well as a maximum number of inner iterations $Q \in \mathbb{N}$, solve*

$$\underset{\theta \in \Theta}{\text{minimize }} \mathcal{U}^{(Q)}(\theta), \quad \text{where} \quad \begin{cases} u^{(0)}(\theta) \text{ is chosen arbitrarily} \\ \text{for } q = 0, 1, \ldots, Q-1 \\ \quad \lfloor \ u^{(q+1)}(\theta) = \mathcal{A}(u^{(q)}(\theta), \theta) \\ w^{(Q)}(\theta) = \mathcal{B}(u^{(Q)}(\theta), \theta), \\ \mathcal{U}^{(Q)}(\theta) = C(w^{(Q)}(\theta)). \end{cases} \tag{6}$$

**Remark 2.2.** Problem 2.3 encompasses many situations encountered in practice. For example, when $\mathcal{B} = \text{Id}$, it reduces to the usual case where $w^{(q+1)}(\theta) = \mathcal{A}(w^{(q)}(\theta), \theta)$. In addition, this formulation also covers dual algorithms: in this case $\mathcal{A}$ corresponds to the dual variable update, and $\mathcal{B}$ denotes the primal-dual relationship (see, e.g., [3]).

The following theorem gives the conditions under which the approximate problem converges to the exact one as the number of inner iterations $Q$ grows.

**Theorem 2.1** (Convergence of the Approximate Problem). *In addition to the assumptions of Problem 2.3, suppose that the iterates $\{w^{(Q)}(\theta)\}_{Q\in\mathbb{N}}$ converge to $\hat{w}(\theta)$ uniformly on $\Theta$ as $Q \to +\infty$. Then the approximate Problem 2.3 converges to the exact Problem 2.2 in the following sense*

$$\inf_{\theta\in\Theta} \mathcal{U}^{(Q)}(\theta) \underset{Q\to+\infty}{\longrightarrow} \inf_{\theta\in\Theta} \mathcal{U}(\theta) \quad and \quad \operatorname*{argmin}_{\theta\in\Theta} \mathcal{U}^{(Q)}(\theta) \underset{Q\to+\infty}{\longrightarrow} \operatorname*{argmin}_{\theta\in\Theta} \mathcal{U}(\theta), \tag{7}$$

*where the latter convergence is meant as set convergence, i.e., for every sequence $(\hat{\theta}^{(Q)})_{Q\in\mathbb{N}}$ such that $\hat{\theta}^{(Q)} \in \operatorname{argmin} \mathcal{U}^{(Q)}$, we have $\operatorname{dist}(\hat{\theta}^{(Q)}, \operatorname{argmin}\mathcal{U}) \to 0$ as $Q \to +\infty$, which is equivalent to $\max\{\operatorname{dist}(\hat{\theta}, \operatorname{argmin}\mathcal{U}) \,|\, \hat{\theta} \in \operatorname{argmin} \mathcal{U}^{(Q)}\} \to 0$ as $Q \to +\infty$.*

Theorem 2.1 justifies the minimization of $\mathcal{U}^{(Q)}$ (for sufficiently large $Q$) instead of $\mathcal{U}$.

## 3 Algorithmic Solution

The lower level problem in (4)-(5), can be, in principle, addressed by several available solvers. However, since this problem is nonsmooth, these solvers are usually nonsmooth as well, that is $\mathcal{A}$ and $\mathcal{B}$ in (6) are nonsmooth. This causes $\mathcal{U}^{(Q)}$ to be nonsmooth, besides being nonconvex. In that case, minimizing $\mathcal{U}^{(Q)}$ is a challenge. Indeed, even just determining a (hyper)subgradient of $\mathcal{U}^{(Q)}$ in a stable fashion by recursively computing a subgradient of $u^{(q)}(\theta)$ might be hopeless. Therefore, we embrace the idea proposed in [24] to devise a smooth algorithm by relying on Bregman proximity operators and we make two advances. First, we propose a new algorithm based on a dual forward-backward scheme with Bregman distances where $\mathcal{A}$ and $\mathcal{B}$ are smooth. Second, by relying on [2], we prove the uniform convergence of such algorithm to the solution of the lower level problem, so to meet the requirements of Theorem 2.1. This approach finally gives a smooth function $\mathcal{U}^{(Q)}$ whose gradient can be recursively computed by applying the standard chain rule [13].

### 3.1 Principle

Since the proposed bilevel problem is a nonconvex problem with possibly many minima, finding the global optimum is out of reach. However, local minima can still be of high quality, meaning that no improvements in the objective can be obtained by small perturbations and that the corresponding objective value is close to the infimum. Let us remark that, since in the parametrization of the groups the ordering is not relevant, the upper level objective function is invariant under permutations of $(\theta_1, \ldots, \theta_L)$, so there are $L!$ equivalent solutions.

In order to solve the bilevel problem, we rely on the following projected gradient descent algorithm

$$(\forall k \in \{0, \ldots, K-1\}), \quad \theta^{(k+1)} = \mathcal{P}_\Theta\big(\theta^{(k)} - \gamma \nabla \mathcal{U}^{(Q)}(\theta^{(k)})\big), \tag{8}$$

where $\mathcal{P}_\Theta$ denotes the projection onto $\Theta$ (see [8] for an efficient projection method when $\Theta$ is the unit simplex) and $\gamma > 0$ is a given step-size. Overall, this procedure requires to compute the $Q$-th iterate $w^{(Q)}(\theta^{(k)})$ as well as the hypergradient $\nabla \mathcal{U}^{(Q)}(\theta^{(k)})$.

Since both the lower and upper level problems are separable with respect to the tasks, the hypergradient is the sum of $T$ terms. In Section 3.4, we design a stochastic variant of (8) taking advantage of this structure.

### 3.2 Solving the Lower Level Problem

In this section, we address the lower level problem in (4)-(5). Since it is is separable with respect to the tasks, without loss of generality we can deal with a single task omitting the index $t$.

**Problem 3.1.** *Given some vectors of outputs $y \in \mathbb{R}^N$, a design matrix $X \in \mathbb{R}^{N\times P}$, regularization parameters $\lambda > 0$ and $\epsilon > 0$, as well as some group structure $\theta \in \Theta$, find*

$$\hat{w}(\theta) = \operatorname*{argmin}_{w\in\mathbb{R}^P} \Big\{ \mathcal{L}(w,\theta) := \underbrace{\frac{1}{2}\|y - Xw\|_2^2 + \frac{\epsilon}{2}\|w\|_2^2}_{f(w)} + \underbrace{\lambda \sum_{l=1}^{L} \|\psi(\theta_l) \odot w\|_2}_{g(A_\theta w)} \Big\}, \tag{9}$$

where $f \in \Gamma_0(\mathbb{R}^P)$ is smooth and $\epsilon$-strongly convex, $g \in \Gamma_0(\mathbb{R}^{P \times L})$ is nonsmooth and $A_\theta$ is the linear operator defined as $A_\theta : w \in \mathbb{R}^P \mapsto (\psi(\theta_1) \odot w, \ldots, \psi(\theta_L) \odot w) \in \mathbb{R}^{P \times L}$.

Let us note that in order to solve Problem 3.1 we cannot use the standard forward-backward algorithm [6, 7] since the proximity operator of $g \circ A_\theta$ cannot be computed in closed form. Moreover, we also ask for a smooth algorithm, meaning one for which $\mathcal{A}$ and $\mathcal{B}$ in (6) are smooth. Therefore, we tackle the dual of Problem 3.1.

**Problem 3.2.** *Find a solution $\hat{u}(\theta)$ of*

$$\underset{u \in \mathbb{R}^{P \times L}}{\text{minimize}} \, f^*(-A_\theta^\top u) + g^*(u), \tag{10}$$

*where $f^*$ and $g^*$ denote the Fenchel conjugates of $f$ and $g$ respectively, and where $A_\theta^\top$ is the transpose of the operator $A_\theta$, that is, $A_\theta^\top : u \in \mathbb{R}^{P \times L} \mapsto \sum_{l=1}^L \psi(\theta_l) \odot u_l \in \mathbb{R}^P$.*

Note that the dual Problem 3.2 admits a solution, since strong duality holds and the primal Problem 3.1 has solutions [3]. Moreover it is a smooth constrained convex optimization problem. Indeed, since $f$ is closed and $\epsilon$-strongly convex, it follows that $f^*$ is everywhere differentiable with $\epsilon^{-1}$-Lipschitz continuous gradient and hence $\nabla[f^* \circ (-A_\theta^\top)] = -A_\theta \nabla f^* \circ (-A_\theta^\top)$ is $\|A_\theta\|^2 \epsilon^{-1}$-Lipschitz continuous. Besides, we have $\nabla f^* = (\nabla f)^{-1} = (X^\top X + \epsilon \mathrm{Id}_P)^{-1}(\cdot + X^\top y)$. On the other hand, $g^*$ is the indicator function of the product of $L$ balls $\mathcal{B}_2(\lambda) \times \ldots \mathcal{B}_2(\lambda) := \mathcal{B}_2(\lambda)^L$, i.e., $g^*(u) = \sum_{l=1}^L \imath_{\mathcal{B}_2(\lambda)}(u_l)$, where $\mathcal{B}_2(\lambda)$ is the closed ball of $\mathbb{R}^P$ centered at zero and of radius $\lambda$.

We propose to solve Problem 3.1 by applying a forward-backward algorithm with Bregman distances to the dual Problem 3.2 [2, 28] and using the primal-dual link $w = \nabla f^*(-A_\theta^\top u)$. This algorithm calls for a Bregman proximity operator of $g^*$ which can be made smooth with an appropriate choice of the Bregman distance. In the following, we provide the related details.

**Definition 3.1** (Bregman Proximity Operator [28]). Let $\mathcal{X}$ be an Euclidean space, $h \in \Gamma_0(\mathcal{X})$ and let $\Phi \in \Gamma_0(\mathcal{X})$ be a Legendre function. Then, the Bregman proximity operator (in Van Nguyen sense) of $h$ with respect to $\Phi$ is

$$\mathrm{prox}_h^\Phi(v) = \underset{u \in \mathcal{X}}{\mathrm{argmin}} \, h(u) + \Phi(u) - \langle u, v \rangle. \tag{11}$$

The dual forward-backward algorithm with Bregman distances (FBB) for Problem 3.1 is as follows. Given some step-size $\gamma > 0$ and $u^{(0)}(\theta)$, then

$$\begin{cases} \text{for } q = 0, 1, \ldots, Q-1 \\ \quad \left\lfloor u^{(q+1)}(\theta) = \mathrm{prox}_{\gamma g^*}^\Phi \left( \nabla \Phi(u^{(q)}(\theta)) + \gamma A_\theta \nabla f^*(-A_\theta^\top u^{(q)}(\theta)) \right) \right. \\ w^{(Q)}(\theta) = \nabla f^*(-A_\theta^\top u^{(Q)}(\theta)). \end{cases} \tag{12}$$

The updating rules in (12) define the mappings $\mathcal{A}$ and $\mathcal{B}$ in Problem 2.3. We note that in this case, since $\nabla f^*$ is an affine mapping, $\mathcal{B}$ is smooth. Whereas the smoothness of $\mathcal{A}$ depends on the choice of the Legendre function $\Phi$. We consider $\Phi(u) = \sum_{l=1}^L \phi(u_l)$ with $\mathrm{dom}\, \phi = \mathcal{B}_2(\lambda)$, so that, for every $l \in \{1, \ldots, L\}$,

$$\begin{aligned} u_l^{(q+1)}(\theta) &= \mathrm{prox}_{\gamma \imath_{\mathcal{B}_2(\lambda)}}^\phi \left( \nabla \phi(u_l^{(q)}(\theta)) + \gamma \psi(\theta_l) \odot \nabla f^*(-A_\theta^\top u^{(q)}(\theta)) \right) \\ &= \underset{u \in \mathbb{R}^P}{\mathrm{argmin}} \, \imath_{\mathcal{B}_2(\lambda)}(u) + \phi(u) - \langle u, \nabla \phi(u_l^{(q)}(\theta)) + \gamma \psi(\theta_l) \odot \nabla f^*(-A_\theta^\top u^{(q)}(\theta)) \rangle \\ &= \underset{u \in \mathbb{R}^P}{\mathrm{argmin}} \, \phi(u) - \langle u, \nabla \phi(u_l^{(q)}(\theta)) + \gamma \psi(\theta_l) \odot \nabla f^*(-A_\theta^\top u^{(q)}(\theta)) \rangle \\ &= \nabla \phi^*(\nabla \phi(u_l^{(q)}(\theta)) + \gamma \psi(\theta_l) \odot \nabla f^*(-A_\theta^\top u^{(q)}(\theta))). \end{aligned} \tag{13}$$

Therefore, in order to make $\mathcal{A}$ smooth, we need to choose the Legendre function $\phi$ so that $\phi^*$ is twice differentiable. Here, we propose to resort to the following function.

**Definition 3.2** (Separable Hellinger-like Function [2]). The separable Hellinger-like function is defined as $\Phi(u) = \sum_{l=1}^L \phi(u_l)$ where for every $u_l \in \mathbb{R}^P$,

$$\phi(u_l) = \begin{cases} -\sqrt{\lambda^2 - \|u_l\|_2^2}, & \text{if} \quad u_l \in \mathcal{B}_2(\lambda), \\ +\infty, & \text{otherwise.} \end{cases} \tag{14}$$

---

**Algorithm 1** Dual forward-backward with Bregman distances: **FBB-GLasso**$(y, X, \lambda, \theta)$

---

**Require:** Data $y$, design matrix $X$, regularization parameter $\lambda$ and group-structure $\theta$
  Set the number of iterations $Q \in \mathbb{N}$ and the step-size $\gamma < \epsilon\lambda^{-1}\|A_\theta\|^{-2}$.
  Set $L$ to the number of groups in $\theta$
  Initialize $u^{(0)}(\theta) \equiv 0_{P \times L}$
  **for** $q = 0$ to $Q - 1$ **do**
    $w^{(q)}(\theta) = (X^\top X + \epsilon\mathrm{Id}_P)^{-1}\big(X^\top y - \sum_{l=1}^{L} \psi(\theta_l) \odot u_l^{(q)}(\theta)\big)$
    **for** $l = 1$ to $L$ **do**
      $v_l^{(q+1)}(\theta) = \frac{u_l^{(q)}(\theta)}{\sqrt{\lambda^2 - \|u_l^q(\theta)\|^2}} + \gamma\psi(\theta_l) \odot w^{(q)}(\theta)$
      $u_l^{(q+1)}(\theta) = \frac{\lambda}{\sqrt{1 + \|v_l^{(q+1)}(\theta)\|_2^2}} v_l^{(q+1)}(\theta)$
    **end for**
  **end for**
  $w^{(Q)}(\theta) = (X^\top X + \epsilon\mathrm{Id}_P)^{-1}\big(X^\top y - \sum_{l=1}^{L} \psi(\theta_l) \odot u_l^{(Q)}(\theta)\big)$
**output** $w^{(Q)}$

---

For such choice, we have that for every $v \in \mathbb{R}^P$, $\nabla\phi^*(v) = \lambda v/\sqrt{1 + \|v\|_2^2}$. The corresponding forward-backward scheme with Bregman distance is given in Algorithm 1 where, for the sake of readability, we introduced the primal iterates $w^{(q)}(\theta)$ and the auxiliary variables $v_l^{(q)}(\theta)$, denoting the argument of $\nabla\phi^*$ in (13). The following theorem addresses the convergence of Algorithm 1. The corresponding proof is given in the supplementary material (Section A.2).

**Theorem 3.1** (Convergence of the Dual FBB Scheme). *The sequence $\{w^{(Q)}(\theta)\}_{Q \in \mathbb{N}}$ generated by Algorithm 1 converges to the solution $\hat{w}(\theta)$ of Problem 3.1 for any step-size $0 < \gamma < \epsilon\lambda^{-1}\|A_\theta\|^{-2}$. In addition if $\gamma = \epsilon\lambda^{-1}\|A_\theta\|^{-2}/2$, then*

$$(\forall Q \in \mathbb{N}) \qquad \frac{1}{2}\|w^{(Q)}(\theta) - \hat{w}(\theta)\|_2^2 \leq \frac{2\lambda\epsilon^{-2}}{Q}\|A_\theta\|^2 D_\Phi(\hat{u}(\theta), u^{(0)}), \tag{15}$$

*where $D_\Phi$ is the Bregman distance associated to $\Phi$, i.e.,*

$$(\forall u \in \mathrm{dom}\,\Phi, \, \forall v \in \mathrm{int}\,\mathrm{dom}\,\Phi), \quad D_\Phi(u, v) = \Phi(u) - \Phi(v) - \langle\nabla\Phi(v), u - v\rangle. \tag{16}$$

**Remark 3.1.** Since $\mathrm{ran}(\psi) \subset [0, 1]$, $\|A_\theta\|^2 \leq L$. If $\psi^2 \leq \xi$, then $\|A_\theta\| \leq 1$; equality is obtained when $\psi^2 = \xi$. Therefore, since $D_\Phi(\cdot, u^{(0)})$ is continuous on $\mathrm{dom}\,\Phi = \mathcal{B}(\lambda)^L$, $\|A_\theta\|^2 D_\Phi(\hat{u}(\theta), u^{(0)})$ in (15) can be uniformly bounded from above on $\Theta$.

Theorem 3.1 and Remark 3.1 establish that $\{w^{(Q)}(\theta)\}_{Q \in \mathbb{N}}$ converges to $\hat{w}(\theta)$ uniformly on $\Theta$ as $Q \to +\infty$ with a sublinear rate. This result applies to every task of the lower level objective in Problem 2.2 and hence it also applies to the collection of tasks $\{\mathrm{w}^{(Q)}(\theta)\}_{Q \in \mathbb{N}}$ and $\hat{w}(\theta)$. Therefore, the requirements of Theorem 2.1 are met and the solutions of Problems 2.3 converge to the solutions of Problem 2.2 as $Q \to +\infty$.

### 3.3 Computing the Hypergradient

In this section, we discuss the computation of the (hyper)gradient of $\mathcal{U}^{(Q)}$. It follows from (6) that, for every $\theta \in \Theta$,

$$\nabla\mathcal{U}^{(Q)}(\theta) = \frac{1}{T}\sum_{t=1}^{T} \underbrace{[(w_t^{(Q)})'(\theta)]^\top}_{\mathbb{R}^{(P \times L) \times P}} \underbrace{\nabla C_t(w_t^{(Q)}(\theta))}_{\mathbb{R}^P} \in \mathbb{R}^{P \times L}, \tag{17}$$

where $\nabla C_t(w_t^{(Q)}(\theta)) = X_t^{(\mathrm{val})\top}(X_t^{(\mathrm{val})}w_t^{(Q)}(\theta) - y_t^{(\mathrm{val})})$. In equation (17) the derivative $(w_t^{(Q)})'(\theta)$ can be computed by recursively differentiating the formulas in (12). This is the so-called *forward mode* for the computation of the hypergradient. However in our setting, this requires storing the derivatives $(u^{(q)})'(\theta)$ which have size $(P \times L) \times (P \times L)$. Here, since we are interested in the product $[(w_t^{(Q)})'(\theta)]^\top \nabla C_t(w_t^{(Q)}(\theta))$ we implement the *reverse mode* differentiation [13] (see also

[11]) which gives a more efficient procedure that only requires storing matrices of size $P \times L$. The details are given in Algorithm 2 in the supplementary material (Section B.1). Finally, as suggested in [13, Chapter 15] and more recently in [24], we implement a variant of Algorithm 2 in which all the derivatives of the mapping $\mathcal{A}$ are evaluated at the last iterate $u^{(Q)}$ (instead of varying during the iterations), This reduces the execution time and memory requirements. In our experiments, we observe that the hypergradient is left unchanged by this operation as long as $Q$ is large enough.

### 3.4 Solving the Approximate Bilevel Problem

Since the hypergradient in (17) has the form of a sum of $T$ terms, each one depending on a single task, we implement a stochastic solver, by estimating the hypergradient $\nabla \mathcal{U}^{(Q)}$ on a single task chosen at random. Here, we resort to the proxSAGA algorithm [26], which is a nonconvex proximal variant of SAGA [9]. The details are given in the supplementary material (Section B.2). In the following we provide the related convergence theorem.

**Theorem 3.2** (Convergence of the Proposed Bilevel Scheme). *Let $\beta$ be the Lipschitz constant of $\nabla \mathcal{U}^{(Q)}$ and $\gamma \leq 1/(5\beta T)$. Let $\{\theta^{(k)}\}_{k=1}^{K}$ be generated according to Algorithm 4 in the supplementary material (Section B.2). Then, for $\tilde{k}$ uniformly sampled from $\{1, \ldots, K\}$, the following holds:*

$$\mathbb{E}\left[\|G_\gamma(\theta^{(\tilde{k})})\|^2\right] \leq \frac{50\beta T^2}{5T - 2} \frac{\mathcal{U}^{(Q)}(\theta^{(0)}) - \mathcal{U}^{(Q)}(\theta^*)}{K}, \tag{18}$$

*where $\theta^*$ is a minimizer of $\mathcal{U}^{(Q)}$ and $G_\gamma$ is the gradient mapping*

$$G_\gamma(\theta) = \frac{1}{\gamma}\left(\theta - \mathcal{P}_\Theta(\theta - \gamma \nabla \mathcal{U}^{(Q)}(\theta))\right). \tag{19}$$

We note that computing the Lipschitz constant $\beta$ is out of reach. Thus, we choose $\gamma$ small enough such that the algorithm converges.

Finally, we suggest the initialization $\theta^{(0)} = \mathcal{P}_\Theta(L^{-1}\mathbb{1}_{P \times L} + n)$ where $n \sim \mathcal{N}(0_{P \times L}, 0.1L^{-1}\mathbb{1}_{P \times L})$ in order to be as uninformative as possible regarding the group structure while still breaking the symmetry by adding a small perturbation.

## 4 Numerical Experiments

In this section, we first devise synthetic experiments to illustrate and assess the performance of the proposed method. Then, we tackle a real-data experiment in the context of gene expression analysis. A MATLAB® toolbox is available upon request to the authors.

### 4.1 Synthetic Experiments

**Experimental setting.** We consider the setting where $N = 50$, $P = 100$, and the group structure $\theta^*$ is made of $L^* = 10$ groups equally distributed over the features such that ($\forall p \in \{1, \ldots, P\}$, $\forall l \in \{1, \ldots, L^*\}$), $\theta^*_{p,l} = 1$ if $p \in \mathcal{G}_l := \{1 + (l-1)(P/L^*), \ldots, l(P/L^*)\}$ and 0 otherwise. In addition, if not stated otherwise, we fix $T = 500$ tasks and every regressor $w_t^*$ is set to have non-zero coefficients equal to 1 in at most 2 groups chosen at random. Both the training, validation and test sets are synthesized as follow. For every task $t \in \{1, \ldots, T\}$, the design matrix $X_t \in \mathbb{R}^{N \times P}$ is first drawn from a standard normal distribution $\mathcal{N}(0_{N \times P}, \mathbb{1}_{N \times P})$ and then normalized column-wise. Finally we define the vector of outputs $y_t = X_t w_t^* + n$ where $n \sim \mathcal{N}(0_N, 0.3\mathbb{1}_N)$.
 We consider the convex relaxation pointed in Remark 2.1, set ($Q = 500, \epsilon = 10^{-3}, \gamma = 0.1, K = 2000$) and denote the proposed solution as $\theta^{\mathrm{BiGL}}$. We also consider its threshold counterpart $\theta^{\mathrm{BiGLThr}}$ where each feature is assigned to its most dominant group. These two solutions are compared with Lasso and oracle Group Lasso, computed respectively for $\theta^{\mathrm{Lasso}} = \mathrm{Id}_P$ and $\theta^{\mathrm{GL}} = \theta^*$. In addition to the validation error, performance are quantified in terms of test and estimation error, $(1/2T)\sum_{t=1}^{T} \|y_t^{(\mathrm{test})} - X_t^{(\mathrm{test})} \cdot \|_2^2$ and $(1/2T)\sum_{t=1}^{T} \|w_t^* - \cdot\|_2^2$ respectively.

**Illustration of the method.** First, we illustrate the well-behaviour of the algorithmic solution, for various values of $\lambda$, when $L^*$ is known. We consider the previously mentioned setting and display in Fig.5 (in the supplementary material) the corresponding oracle $w^*$ (top left) exhibiting 10 groups.

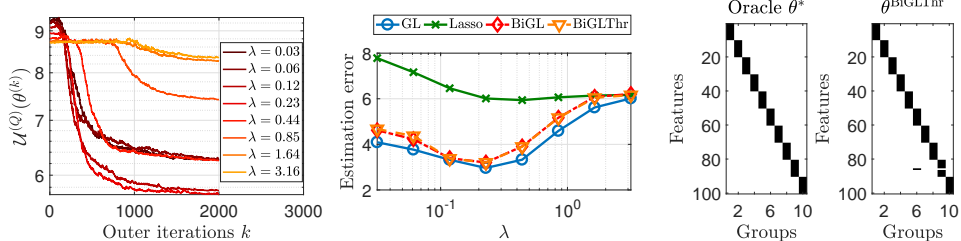

Figure 1: The minimization of $\mathcal{U}^{(Q)}$ is displayed in the left plot for various $\lambda$. Comparison of estimation errors (middle) show that the proposed BiGL and BiGLThr estimates yield performance close to the oracle GL. In addition, $\theta^{\text{BiGLThr}}$ satisfactorily agrees with the oracle $\theta^*$ (right).

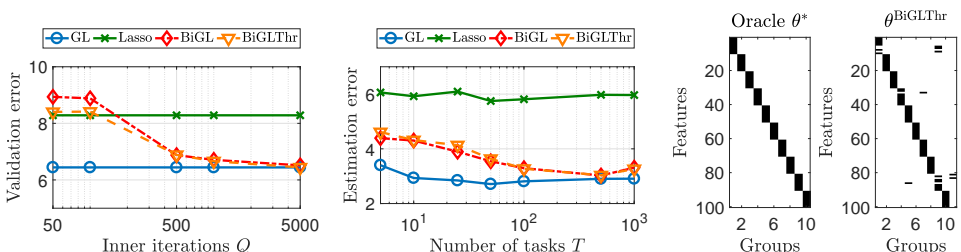

Figure 2: Left and middle plots illustrate the impact of $Q$ and $T$ on the validation and the estimation error respectively. The right-hand figure shows that an adequate estimation of the groups can be obtained even when the number of groups is set to 20 instead of 10.

Figure 1 (left) shows, for several values of $\lambda$, how the upper level objective decreases as the number of outer iterations $k$ grows. Even though convergence is not yet fully reached, the corresponding solutions still yield performance close to oracle Group Lasso as shown by the validation, test and estimation error (see Figure 1 and supplementary material). More importantly, Figure 1 (right) shows that for the $\lambda$ minimizing the validation error, denoted $\lambda_{\min}$, the corresponding estimated groups $\theta^{\text{BiGLThr}}$ satisfactorily agree with the oracle $\theta^*$, thus confirming that minimizing the validation error is an adequate way to learn the groups.

**Impact of the number of inner iterations $Q$.** Now that a proof of concept has been provided, we propose to investigate the impact of $Q$ on the validation error. To do so, we repeat the same experiment for $\lambda = \lambda_{\min}$ and different values of $Q$. Once the estimates $\theta^{\text{BiGL}}$ and $\theta^{\text{BiGLThr}}$ are obtained, the validation error (where the $\hat{w}(\cdot)$'s are computed *a posteriori* for $10^4$ iterations) for each of the four methods is plotted as a function of $Q$ in Fig.2 (left). The results show that increasing $Q$ sufficiently permits reaching performance close to GL. In addition, we stress that, for $Q \geq 500$, the performance of BiGL and BiGLThr become indistinguishable, thus showing that the algorithm does tend to assign a single group to each feature.

**Impact of the number of tasks $T$.** Here we investigate how the estimation errors varies as the number of task $T$ increases, see in Fig.2 (middle). While the performance of Lasso and GL do not significantly depend on $T$, we observe that the performance of BiGL and BiGLThr get close to those of GL as $T$ grows. Similar conclusions can be drawn regarding the test error. Hence, this confirms that learning the groups is intrinsically a multi-task problem that benefits from having a large number of tasks.

**Impact of the number of groups $L$.** While in the previous experiments the number of groups was known *a priori* ($L^* = 10$), here we relax this assumption and let the algorithm find at most $L = 20$ groups. We repeat the experiment and show the results in Fig. 2 (right). Note that 9 out of the 10 extra groups are not displayed since they were found empty, while the remaining group contains very few features. Overall, the oracle $\theta^*$ is still satisfactorily estimated.

**Impact of groups sizes.** We repeat the same experiment except that $\theta^*$ is now made of 5 groups of 5 features and 5 groups of 15 features. The proposed method still satisfactorily estimates groups of different size, as Figure 3 shows, in case $L^*$ is known as well as if $L^*$ is overestimated ($L = 20$).

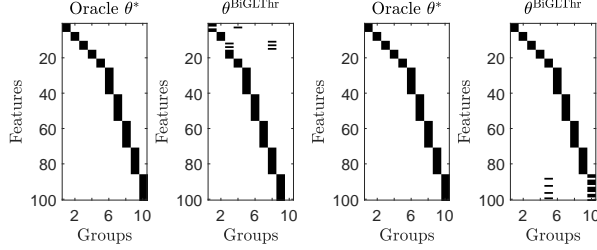

Figure 3: Illustration of the estimated group-structure when the oracle groups have different sizes. Left: initialization with the correct number of groups $L = 10$. Right: initialization with an overestimation $L = 20$ of the number of groups.

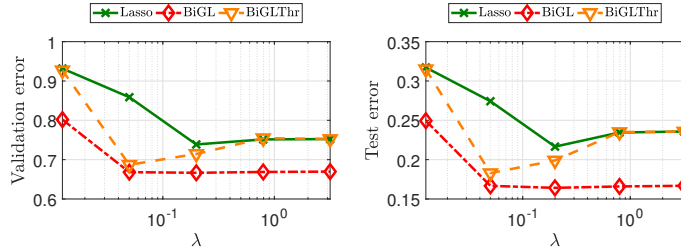

Figure 4: Application to the prediction of gene ontology classes from regulatory motifs. Our approach is able to reach a lower prediction error than Lasso by partitioning the features into 30 groups.

## 4.2 Application to Real Data

Understanding the complexity of gene expression networks and the mechanisms involved in its regulation constitutes an extremely difficult task [25]. In this section, we lead a preliminary experiment on gene expression data collected from `https://www.ensembl.org/` using BioMart. The data consists of $N = 60$ genes each one characterized by $P = 50$ features, corresponding to the regulatory motifs in promoters. These samples may belong to at most 108 gene ontology classes. Each class corresponds to a very specific molecular function of the transcripts. The data set is split into training, validation and test sets of 20 genes each. We perform a multi-task classification ($T = 108$) where each task consists of a one versus all classification problem. Our bilevel algorithm is initialized with $L = 50$ possible groups. Validation and test errors are displayed in Fig. 4 as functions of $\lambda$. The results show a significant decrease in prediction error when using the proposed method compared with Lasso. In addition, $\theta^{\mathrm{BiGLThr}}$ suggests that there exist 30 relevant groups among the features. This preliminary experiment is encouraging and paves the way to set extended and more comprehensive experiments in gene data analysis.

## 5 Conclusion

This contribution studied the problem of learning the groups by solving a continuous bilevel problem. We replaced the exact Group Lasso optimization problem by a smooth dual forward-backward algorithm with Bregman distances. This method is in the line of what has been proposed in [24]. We also provided theoretical justifications of this approximation method which, to the best of our knowledge, is new. When compared to standard sparse regression methods, the proposed procedure achieved equivalent performance to the oracle Group Lasso where the true groups are known. Moreover, when the numbers of tasks and inner iterations are sufficiently large, a satisfactory estimate of the groups can be obtained even if the number of groups are unknown. One of the advantages of the proposed approach is that it can be easily adapted to different convex losses with Lipschitz-continuous gradient. Future works notably include the extension to overlapping groups [15] and could also aim at learning the groups in group-sparse classification problems.

**Acknowledgments**

We wish to thank Luca Franceschi and the anonymous referees for their useful comments. We also would like to thank Giorgio Valentini for providing the gene expression dataset. This work was supported in part by SAP SE.

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
