[Supplementary Material]

# Supplementary Material to "Bilevel Learning of the Group Lasso Structure"

Section A collects the proofs of the main results presented in the paper. Full details on the proposed algorithms are given in Section B. Finally, Section C contains additional results on synthetic data.

## A Proofs

### A.1 Analysis of the Bilevel Framework

**Proof. of Proposition 2.1.** We first note that $\theta \mapsto \hat{w}(\theta)$ is bounded. Indeed it follows from the definition of $\mathcal{L}$ in (5) that $(\epsilon/(2T))\|\hat{w}(\theta)\|^2 \leq \mathcal{L}(w(\theta), \theta) \leq \mathcal{L}(0, \theta) = (1/(2T)) \sum_{t=1}^{T} \|y_t\|^2$. Now, we show that $\theta \mapsto \hat{w}(\theta)$ is continuous. Let $\bar{\theta} \in \Theta$ and let $(\theta^{(n)})_{n \in \mathbb{N}}$ be a sequence in $\Theta$ such that $\theta^{(n)} \to \bar{\theta}$. Since $(\hat{w}(\theta^{(n)}))_{n \in \mathbb{N}}$ is bounded, in order to show that $\hat{w}(\theta^{(n)}) \to \hat{w}(\bar{\theta})$, it is sufficient to prove that $\hat{w}(\bar{\theta})$ is the unique cluster point of $(\hat{w}(\theta^{(n)}))_{n \in \mathbb{N}}$. So, let $\hat{w}(\theta^{(\kappa_n)})_{n \in \mathbb{N}}$ be a converging subsequence, say to $\bar{w}$. Then, since $\mathcal{L}$ is jointly continuous,

$$\forall w \in \mathbb{R}^{P \times T} \quad \mathcal{L}(\bar{w}, \bar{\theta}) = \lim_{n \to +\infty} \mathcal{L}(\hat{w}(\theta^{(\kappa_n)}), \theta^{(\kappa_n)}) \leq \lim_{n \to +\infty} \mathcal{L}(w, \theta^{(\kappa_n)}) = \mathcal{L}(w, \bar{\theta}). \quad (1)$$

Therefore, $\bar{w} = \operatorname{argmin}_{w \in \mathbb{R}^{P \times T}} \mathcal{L}(w, \bar{\theta}) = \hat{w}(\bar{\theta})$. Thus, $\mathcal{U}: \theta \in \Theta \to C(\hat{w}(\theta)) \in \mathbb{R}$ is continuous and hence, since $\Theta$ is compact, it has a minimizer. □

**Proof. of Theorem 2.1.** We first prove that $\mathcal{U}^{(Q)}(\theta) \to \mathcal{U}(\theta)$ uniformly on $\Theta$ as $Q \to +\infty$. Indeed let $\varepsilon > 0$. Then, since $C$ is uniformy continuous on $\Theta$, there exists $\delta > 0$ such that

$$\forall w, w' \in \mathbb{R}^{P \times T}, \quad \|w - w'\| \leq \delta \implies |C(w) - C(w')| \leq \varepsilon. \quad (2)$$

Since $w^{(Q)}(\theta) \to \hat{w}(\theta)$ uniformly on $\Theta$ as $Q \to +\infty$, there exists $K \in \mathbb{N}$ such that for every integer $Q \geq K$, $\sup_{\theta \in \Theta} \|w^{(Q)}(\theta) - \hat{w}(\theta)\| \leq \delta$ and hence $\sup_{\theta \in \Theta} \|C(w^{(Q)}(\theta)) - C(\hat{w}(\theta))\| \leq \varepsilon$.

Now, let $(\hat{\theta}^{(Q)})_{Q \in \mathbb{N}}$ be a sequence in $\Theta$ such that, for every $Q \in \mathbb{N}$, $\hat{\theta}^{(Q)} \in \operatorname{argmin} \mathcal{U}^{(Q)}$ We prove that

(i) $(\hat{\theta}^{(Q)})_{Q \in \mathbb{N}}$ admits a convergent subsequence.

(ii) for every subsequence $(\hat{\theta}^{(K_Q)})_{Q \in \mathbb{N}}$ such that $\hat{\theta}^{(K_Q)} \to \bar{\theta}$ as $Q \to +\infty$, we have $\bar{\theta} \in \operatorname{argmin} \mathcal{U}$ and $\mathcal{U}_{K_Q}(\hat{\theta}^{(K_Q)}) \to \inf \mathcal{U}$ as $Q \to +\infty$.

(iii) $\inf \mathcal{U}^{(Q)} \to \inf \mathcal{U}$ as $Q \to +\infty$.

(iv) $\operatorname{dist}(\hat{\theta}^{(Q)}, \operatorname{argmin} \mathcal{U}) \to 0$ as $Q \to +\infty$.

The first point follows from the fact that $\Theta$ is compact.

Concerning the second point, let $(\hat{\theta}_{K_Q})_{Q \in \mathbb{N}}$ be a subsequence such that $\hat{\theta}_{K_Q} \to \bar{\theta}$. Since $\mathcal{U}_{K_Q}$ converges uniformy to $\mathcal{U}$ on $\Theta$ as $Q \to +\infty$, we have

$$|\mathcal{U}_{K_Q}(\hat{\theta}^{(K_Q)}) - \mathcal{U}(\hat{\theta}^{(K_Q)})| \leq \sup_{\theta \in \Theta} |\mathcal{U}_{K_Q}(\theta) - \mathcal{U}(\theta)| \to 0 \quad \text{as } Q \to +\infty.$$

Therefore, using also the continuity of $\mathcal{U}$, we have

$$\forall \theta \in \Theta, \quad \mathcal{U}(\bar{\theta}) = \lim_Q \mathcal{U}(\hat{\theta}^{(K_Q)}) = \lim_Q \mathcal{U}_{K_Q}(\hat{\theta}^{(K_Q)})$$
$$\leq \lim_Q \mathcal{U}_{K_Q}(\theta) = \mathcal{U}(\theta).$$

So, $\bar{\theta} \in \operatorname{argmin}\mathcal{U}$ and $\mathcal{U}(\bar{\theta}) = \lim_Q \mathcal{U}_{K_Q}(\hat{\theta}^{(K_Q)}) \leq \inf\mathcal{U} = \mathcal{U}(\bar{\theta})$, that is, $\lim_Q \mathcal{U}_{K_Q}(\hat{\theta}^{(K_Q)}) = \inf\mathcal{U}$.

As regards the third point, we proceed by contradiction. If $(\mathcal{U}^{(Q)}(\hat{\theta}^{(Q)}))_{Q \in \mathbb{N}}$ does not converge to $\inf\mathcal{U}$, then there exists an $\varepsilon > 0$ and a subsequence $(\mathcal{U}_{K_Q}(\hat{\theta}^{(K_Q)}))_{Q \in \mathbb{N}}$ such that

$$|\mathcal{U}_{K_Q}(\hat{\theta}^{(K_Q)}) - \inf\mathcal{U}| \geq \varepsilon, \quad \forall Q \in \mathbb{N} \tag{3}$$

Now, let $(\hat{\theta}^{(K_Q^{(1)})})_{Q \in \mathbb{N}}$ be a convergent subsequence of $(\hat{\theta}^{(K_Q)})_{Q \in \mathbb{N}}$. Suppose that $\hat{\theta}^{(K_Q^{(1)})} \to \bar{\theta}$. Clearly $(\hat{\theta}^{(K_Q^{(1)})})_{Q \in \mathbb{N}}$ is also a subsequence of $(\hat{\theta}^{(Q)})_{Q \in \mathbb{N}}$. Then, it follows from point (ii) above that $\mathcal{U}_{K_Q^{(1)}}(\hat{\theta}^{(K_Q^{(1)})}) \to \inf\mathcal{U}$. This latter finding together with equation (3) gives a contradiction.

Finally, concerning the last point, we set $a = \limsup_{Q \to +\infty} \operatorname{dist}(\hat{\theta}^{(Q)}, \operatorname{argmin}\mathcal{U}) \in \mathbb{R}_+ \cup \{+\infty\}$. Then there exists a subsequence $(\hat{\theta}^{(K_Q)})_{Q \in \mathbb{N}}$ such that $\operatorname{dist}(\hat{\theta}^{(K_Q)}, \operatorname{argmin}\mathcal{U}) \to a$ as $Q \to +\infty$. Now, since $(\hat{\theta}^{(K_Q)})_{Q \in \mathbb{N}}$ is bounded, it has a subsequence $(\hat{\theta}^{(K_Q^1)})_{Q \in \mathbb{N}}$ such that $\hat{\theta}^{(K_Q^1)} \to \bar{\theta}$ for some $\bar{\theta} \in \Theta$. Moreover, it follows from point (ii) above that $\bar{\theta} \in \operatorname{argmin}\mathcal{U}$. Therefore, since $\operatorname{dist}(\cdot, \operatorname{argmin}\mathcal{U})$ is continuous, we have $a = \lim_{Q \to +\infty} \operatorname{dist}(\hat{\theta}^{(K_Q^1)}, \operatorname{argmin}\mathcal{U}) = \operatorname{dist}(\bar{\theta}, \mathcal{U}) = 0$. $\square$

## A.2 Convergence of the Forward-Backward Scheme with Bregman Distances

In this section, we provide the proof of Theorem 3.1, which will be based on the results in [1]. To that purpose we need some preliminary results.

**Proposition A.1.** *The Legendre function $\Phi$ defined in Definition 3.2 is $\lambda^{-1}$ strongly convex.*

*Proof.* Let $u = (u_1, \ldots, u_L) \in \operatorname{int} \operatorname{dom}\Phi = \operatorname{int}(B_2(\lambda))^L$. Since $\Phi$ is separable, its Hessian $\nabla^2 \Phi(u)$ is block-diagonal and for every $v = (v_1, \ldots, v_L) \in \mathbb{R}^{P \times L}$, we have

$$v^\top \nabla^2 \Phi(u)v = \sum_{l=1}^{L} \frac{\|v_l\|^2}{\sqrt{\lambda^2 - \|u_l\|^2}} + \frac{(v_l^\top u_l)^2}{(\lambda^2 - \|u_l\|^2)^{3/2}} \tag{4}$$

$$\geq \sum_{l=1}^{L} \frac{\|v_l\|^2}{\sqrt{\lambda^2 - \|u_l\|^2}} \tag{5}$$

$$\geq \sum_{l=1}^{L} \frac{1}{\lambda} \|v_l\|_2^2 = \frac{1}{\lambda} \|v\|^2, \tag{6}$$

which completes the proof. $\square$

**Proposition A.2** (Lipschitz-like constant). *Let $\mu = \epsilon^{-1} \lambda \|A_\theta\|^2$. Then the function $\mu\Phi - f^* \circ (-A_\theta^\top)$ is convex.*

*Proof.* The function $\mu\Phi - f^* \circ (-A_\theta^\top)$ is twice continuously differentiable on $\operatorname{int} \operatorname{dom}\Phi$ (which equals to $\operatorname{int}(B_2(\lambda)^L)$. Therefore the statement is equivalent to

$$\forall u \in \operatorname{int} \operatorname{dom}\Phi, \forall v \in \mathbb{R}^{P \times L} \quad \mu v^\top \nabla^2 \Phi(u)v - v^\top \nabla^2[f^* \circ (-A_\theta^\top)](u)v \geq 0. \tag{7}$$

Since the function $f^* \circ (-A_\theta^\top)$ has a Lipschitz continuous gradient with constant $\epsilon^{-1} \|A_\theta\|^2$, we have that $v^\top \nabla^2[f^* \circ (-A_\theta^\top)](u)v \leq \epsilon^{-1} \|A_\theta\|^2 \|v\|^2$. Moreover, it follows from Proposition A.1 that $\Phi$ is $\lambda^{-1}$-strongly convex, hence $\mu v^\top \nabla^2 \Phi(u)v \geq \mu/\lambda \|v\|^2$. Therefore

$$\mu v^\top \nabla^2 \Phi(u)v - v^\top \nabla^2[f^* \circ (-A_\theta^\top)](u)v \geq \left(\frac{\mu}{\lambda} - \frac{\|A_\theta\|^2}{\epsilon}\right) \|v\|^2 \geq 0,$$

and the statement follows. $\square$

**Proposition A.3.** *The symmetry coefficient $\alpha(\Phi)$ of the Legendre function of Definition 3.2, defined as*

$$\alpha(\Phi) = \inf\left\{ \frac{D_\Phi(u, v)}{D_\Phi(v, u)} \,\bigg|\, (u, v) \in \operatorname{int} \operatorname{dom}\Phi \times \operatorname{int} \operatorname{dom}\Phi, u \neq v \right\}, \tag{8}$$

*is equal to zero.*

*Proof.* This follows from the general Proposition 2 in [1], since $\operatorname{dom}\Phi$ is not open. $\qquad\square$

**Proof. of Theorem 3.1** Let $\mathcal{F}$ denote the objective function in (10). It follows from standard argument in convex duality theory (see, e.g., [2]) that for every $u \in \mathbb{R}^{P \times L}$, setting $w = \nabla f^*(-A_\theta^\top u)$, we have

$$\frac{\epsilon}{2}\|w - \hat{w}(\theta)\|^2 \le \mathcal{F}(u) - \inf \mathcal{F}. \tag{9}$$

Then the statement follows from Theorem 1 in [1]. Indeed, in the setting of Problem 3.2, with $\Phi$ as in Definition 3.2, we have $\operatorname{dom}\Phi = B_2(\lambda)^L$ (which is a closed set) and moreover the following conditions are satisfied:

1. *(Well-posedness of the method)* $\operatorname{argmin}_{u \in \overline{\operatorname{dom}\Phi}} \mathcal{F}(u)$ is compact (see Lemma 2 in [1]);
2. *(Lipschitz-like)* There exist a Lipschitz-like constant $\mu > 0$ ($\mu = \epsilon^{-1}\lambda\|A_\theta\|^2$) such that $\mu\Phi - f^* \circ (-A_\theta^\top)$ is convex;
3. *(Step-size condition)* The step-size is such that $0 < \gamma < (1 + \alpha(\Phi))/\mu$, where $\alpha(\Phi)$ is the symmetry coefficient defined in (8).

Therefore, according to Theorem 1 in [1] the following hold

1. *(Monotonicity)* $\{\mathcal{F}(u^{(q)}(\theta))\}_{q \in \mathbb{N}}$ is nonincreasing.
2. *(Convergence in objective values)* $\lim_{q \to +\infty} \mathcal{F}(u^{(q)}(\theta)) = \mathcal{F}(\hat{u}(\theta))$
3. *(Global estimate in objective values)* If $\gamma = (1 + \alpha(\Phi))/(2\mu)$, then

$$(\forall u \in \operatorname{dom}\Phi)(\forall q \in \mathbb{N}) \quad \mathcal{F}(u^{(q)}(\theta)) - \mathcal{F}(u) \le \frac{2\mu}{(1 + \alpha(\Phi))q} D_\Phi(u, u^0(\theta)). \tag{10}$$

The statement follows from (10) (with $u = \hat{u}(\theta)$) and (9). $\qquad\square$

# B Algorithms

In this section, we detail the procedure for computing the hypergradient as well as the entire bilevel algorithm.

## B.1 Reverse Mode Computation of the Hypergradient

Recalling the definitions given in Problem 2.1 we have that

$$\mathcal{U}^{(Q)}(\theta) = \frac{1}{T}\sum_{t=1}^{T} C_t(w_t^{(Q)}(\theta)), \tag{11}$$

where, each task $w_t^{(Q)}(\theta)$ is computed by algorithm (12). Therefore

$$\nabla\mathcal{U}^{(Q)}(\theta) = \frac{1}{T}\sum_{t=1}^{T} \nabla\mathcal{U}_t^{(Q)}(\theta), \qquad \mathcal{U}_t^{(Q)}(\theta) =: C_t(w_t^{(Q)}(\theta)) \tag{12}$$

and the problem is reduced to the computation of the gradient of $\mathcal{U}_t^{(Q)}(\theta)$. Thus, we can deal with a single task and assume that

$$\mathcal{U}^{(Q)}(\theta) = C(w^{(Q)}(\theta)), \tag{13}$$

where $w^{(Q)}(\theta) \in \mathbb{R}^P$ is computed by an algorithm of the following form

$$\begin{cases} u^{(0)}(\theta) \equiv 0 \in \mathbb{R}^{P \times L} \\ \text{for } q = 0, 1, \dots, Q-1 \\ \quad \lfloor\ u^{(q+1)}(\theta) = \mathcal{A}(u^{(q)}(\theta), \theta) \\ w^{(Q)}(\theta) = \mathcal{B}(u^{(Q)}(\theta), \theta), \end{cases} \tag{14}$$

where $\mathcal{A}\colon \mathbb{R}^{P\times L}\times\Theta\to\mathbb{R}^{P\times L}$ and $\mathcal{B}\colon \mathbb{R}^{P\times L}\times\Theta\to\mathbb{R}^{P}$. We denote by $\partial_1\mathcal{A}(u,\theta)$ and $\partial_2\mathcal{A}(u,\theta)$ the partial derivatives of $\mathcal{A}$ with respect to the variable $u$ and $\theta$ respectively. Note that both the partial derivatives are linear operators from $\mathbb{R}^{P\times L}$ to $\mathbb{R}^{P\times L}$. The same notation is used for the partial derivatives of $\mathcal{B}$, which, evaluated at a given point, are linear operators from $\mathbb{R}^{P\times L}$ to $\mathbb{R}^{P}$. Using (13) and the last equation in (14) we get

$$\nabla\mathcal{U}^{(Q)}(\theta) = (u^{(Q)})'(\theta)^{\top}\partial_1\mathcal{B}(u^{(Q)}(\theta),\theta)^{\top}\nabla C(w^{(Q)}(\theta)) + \partial_2\mathcal{B}(u^{(Q)}(\theta),\theta)^{\top}\nabla C(w^{(Q)}(\theta)). \tag{15}$$

Moreover, using the updating rule for $u^{(q)}(\theta)$ in (14) we have

$$(u^{(q+1)})'(\theta) = \partial_1\mathcal{A}(u^{(q)}(\theta),\theta)(u^{(q)})'(\theta) + \partial_2\mathcal{A}(u^{(q)}(\theta),\theta). \tag{16}$$

Setting $A_1^{(q)}(\theta) = \partial_1\mathcal{A}(u^{(q)}(\theta),\theta)$ and $A_2^{(q)}(\theta) = \partial_2\mathcal{A}(u^{(q)}(\theta),\theta)$, we have

$$(u^{(q+1)})'(\theta)^{\top} = (u^{(q)})'(\theta)^{\top}A_1^{(q)}(\theta)^{\top} + A_2^{(q)}(\theta)^{\top}. \tag{17}$$

Then, combining the two equations above we have

$$
\begin{aligned}
\nabla\mathcal{U}^{(Q)}(\theta) &= (u^{(Q)})'(\theta)^{\top}\partial_1\mathcal{B}(u^{(Q)}(\theta),\theta)^{\top}\nabla C(w^{(Q)}(\theta)) + \partial_2\mathcal{B}(u^{(Q)}(\theta),\theta)^{\top}\nabla C(w^{(Q)}(\theta)) \\
&= (u^{(Q-1)})'(\theta)^{\top}A_1^{(Q-1)}(\theta)^{\top}\underbrace{\partial_1\mathcal{B}(u^{(Q)}(\theta),\theta)^{\top}\nabla C(w^{(Q)}(\theta))}_{a_Q} \\
&\quad + A_2^{(Q-1)}(\theta)^{\top}\underbrace{\partial_1\mathcal{B}(u^{(Q)}(\theta),\theta)^{\top}\nabla C(w^{(Q)}(\theta))}_{a_Q} \\
&\quad + \underbrace{\partial_2\mathcal{B}(u^{(Q)}(\theta),\theta)^{\top}\nabla C(w^{(Q)}(\theta))}_{b_Q} \\
&= (u^{(Q-1)})'(\theta)^{\top}\underbrace{A_1^{(Q-1)}(\theta)^{\top}a_Q}_{a_{Q-1}} + \underbrace{A_2^{(Q-1)}(\theta)^{\top}a_Q + b_Q}_{b_{Q-1}} \\
&= (u^{(Q-2)})'(\theta)^{\top}\underbrace{A_1^{(Q-2)}(\theta)^{\top}a_{Q-1}}_{a_{Q-2}} + \underbrace{A_2^{(Q-2)}(\theta)^{\top}a_{Q-1} + b_{Q-1}}_{b_{Q-2}} \\
&= \dots\dots\dots\dots\dots\dots\dots\dots\dots\dots\dots\dots\dots\dots\dots \\
&= \underbrace{A_2^{(0)}(\theta)^{\top}a_1 + b_1}_{b_0},
\end{aligned}
$$

where in the last line we used that $\mathrm{u}^{(0)}(\theta)$ is constant. Therefore, $\nabla\mathcal{U}^{(Q)}$ can be computed by the procedure detailed in Algorithm 2.

We now specialize Algorithm 2 to the case of Group Lasso and algorithm (12). In this case, the update rules are as follows

$$
\begin{aligned}
\mathcal{A}(u,\theta) &= \nabla\Phi^*(\nabla\Phi(u) + \gamma A_\theta\mathcal{B}(u,\theta)) \\
\mathcal{B}(u,\theta) &= \nabla f^*(-A_\theta^{\top}u) \\
&= (X^{\top}X + \epsilon\mathrm{Id}_P)^{-1}(X^{\top}y - A_\theta^{\top}u),
\end{aligned}
$$

where, for every $u = (u_l)_{1\le l\le L} \in \mathbb{R}^{P\times L}$ and $v = (v_l)_{1\le l\le L} \in \mathbb{R}^{P\times L}$ and every $l = 1,\dots,L$,

$$\nabla_l\Phi(u) = \nabla\phi(u_l) = \frac{u_l}{\sqrt{\lambda^2 - \|u_l\|_2^2}} \quad\text{and}\quad \nabla_l\Phi^*(v) = \nabla\phi^*(v_l) = \frac{\lambda v_l}{\sqrt{1 + \|v_l\|_2^2}}. \tag{18}$$

Moreover, for every $a = (a_l)_{1\le l\le L} \in \mathbb{R}^{P\times L}$

$$\nabla^2\Phi(u)[a] = \left(\nabla^2\phi(u_l)[a_l]\right)_{1\le l\le L} = \left(\frac{\langle u_l, a_l\rangle u_l}{\left(\lambda^2 - \|u_l\|_2^2\right)^{3/2}} + \frac{a_l}{\sqrt{\lambda^2 - \|u_l\|_2^2}}\right)_{1\le l\le L} \in \mathbb{R}^{P\times L} \tag{19}$$

---
**Algorithm 2** Hypergradient computation (Reverse mode): **Hypergradient**$(\theta, Q)$
---
**Require:** Group structure $\theta$, number of inner iterations $Q$.
  Initialize $u^{(0)}(\theta) \equiv 0 \in \mathbb{R}^{P \times L}$.
  **for** $q = 1$ to $Q$ **do**
    $u^{(q)}(\theta) = \mathcal{A}(u^{(q-1)}(\theta), \theta)$.
  **end for**
**output** 1. $u^{(0)}(\theta), \ldots, u^{(Q)}(\theta)$, $w^{(Q)}(\theta) = \mathcal{B}(u^Q(\theta), \theta)$.
  Initialize $a_Q = \partial_1 \mathcal{B}(u^{(Q)}(\theta), \theta)^\top \nabla C(x^{(Q)}(\theta))$, $b_Q = \partial_2 \mathcal{B}(u^{(Q)}(\theta), \theta)^\top \nabla C(w^{(Q)}(\theta))$.
  **for** $q = Q - 1$ to $0$ **do**
    $a^{(q)} = \partial_1 \mathcal{A}(u^{(q)}(\theta), \theta)^\top a^{(q+1)}$
    $b^{(q)} = \partial_2 \mathcal{A}(u^{(q)}(\theta), \theta)^\top a^{(q+1)} + b^{(q+1)}$.
  **end for**
**output** 2. Hypergradient $\nabla \mathcal{U}^{(Q)}(\theta) = b^{(0)}$.
---

and

$$\nabla^2 \Phi^*(v)[a] = \left(\nabla^2 \phi^*(v_l)[a_l]\right)_{1 \leq l \leq L} = \left(-\frac{\lambda \langle v_l, a_l \rangle v_l}{\left(1 + \|v_l\|_2^2\right)^{3/2}} + \frac{\lambda a_l}{\sqrt{1 + \|v_l\|_2^2}}\right)_{1 \leq l \leq L} \in \mathbb{R}^{P \times L}.$$
(20)

Note that both $\nabla^2 \Phi(u)$ and $\nabla^2 \Phi^*(v)$ are symmetric linear operators from $\mathbb{R}^{P \times L}$ to $\mathbb{R}^{P \times L}$.

Therefore,

$$\partial_1 \mathcal{A}(u, \theta) = \nabla^2 \Phi^*(\nabla \Phi(u) + \gamma A_\theta \mathcal{B}(u, \theta)) \circ [\nabla^2 \Phi(u) + \gamma A_\theta \partial_1 \mathcal{B}(u, \theta)]$$
$$\partial_2 \mathcal{A}(u, \theta) = \nabla^2 \Phi^*(\nabla \Phi(u) + \gamma A_\theta \mathcal{B}(u, \theta)) \circ [\gamma A_\theta \partial_2 \mathcal{B}(u, \theta) + \gamma A_. \mathcal{B}(u, \theta)]$$

and

$$\partial_1 \mathcal{B}(u, \theta) = -(X^\top X + \epsilon \mathrm{Id}_P)^{-1} A_\theta^\top$$
$$\partial_2 \mathcal{B}(u, \theta) = -(X^\top X + \epsilon \mathrm{Id}_P)^{-1} (A_.^* u).$$

Moreover, since the linear operator $T_x \colon \vartheta \mapsto A_\vartheta x$ (occurring in $\partial_2 \mathcal{A}(u, \theta)$) is symmetric and the adjoint of the linear operator $S_u \colon \theta \mapsto A_\theta^\top u$ (occurring in $\partial_2 \mathcal{B}(u, \theta)$) is $A_u$, we have

$$\begin{cases} \partial_1 \mathcal{A}(u, \theta)^\top = [\nabla^2 \Phi(u) + \gamma \partial_1 \mathcal{B}(u, \theta)^\top A_\theta^\top] \circ \nabla^2 \Phi^*(\nabla \Phi(u) + \gamma A_\theta \mathcal{B}(u, \theta)) \\ \partial_2 \mathcal{A}(u, \theta)^\top = [\gamma \partial_2 \mathcal{B}(u, \theta)^\top A_\theta^* + \gamma A_. \mathcal{B}(u, \theta)] \circ \nabla^2 \Phi^*(\nabla \Phi(u) + \gamma A_\theta \mathcal{B}(u, \theta)) \\ \partial_1 \mathcal{B}(u, \theta)^\top = -A_\theta(X^\top X + \epsilon \mathrm{Id}_P)^{-1} \\ \partial_2 \mathcal{B}(u, \theta)^\top = -A_u(X^\top X + \epsilon \mathrm{Id}_P)^{-1}. \end{cases}$$
(21)

Hence, for every $a \in \mathbb{R}^{P \times L}, \partial_1 \mathcal{A}(u, \theta)^\top a$ and $\partial_2 \mathcal{A}(u, \theta)^\top a$ can be computed as follows:

$$v = \left(\nabla \phi(u_l) + \gamma \theta_l \odot \mathcal{B}(u, \theta)\right)_{1 \leq l \leq L},$$

$$\partial_1 \mathcal{A}(u, \theta)^\top a = \nabla^2 \Phi(u)\left[\nabla^2 \Phi^*(v)[a]\right] + \gamma \partial_1 \mathcal{B}(u, \theta)^\top A_\theta^\top \nabla^2 \Phi^*(v)[a]$$
$$= \left(\nabla^2 \phi(u_l)\left[\nabla^2 \phi^*(v_l)[a_l]\right] - \gamma \theta_l \odot (X^\top X + \epsilon \mathrm{Id}_P)^{-1} \nabla^2 \Phi^*(v)[a]\right)_{1 \leq l \leq L},$$

$$\partial_2 \mathcal{A}(u, \theta)^\top a = \gamma \partial_2 \mathcal{B}(u, \theta)^\top A_\theta^\top \nabla^2 \Phi^*(v)[a] + \gamma A_{\nabla^2 \Phi^*(v)[a]} \mathcal{B}(u, \theta)$$
$$= -\gamma A_u (X^\top X + \epsilon \mathrm{Id}_P)^{-1} A_\theta^\top \nabla^2 \Phi^*(v)[a] + \gamma \left(\nabla^2 \phi^*(v_l)[a_l] \odot \mathcal{B}(u, \theta)\right)_{1 \leq l \leq L}$$
$$= \gamma \left(-u_l \odot (X^\top X + \epsilon \mathrm{Id}_P)^{-1} A_\theta^\top \nabla^2 \Phi^*(v)[a] + \nabla^2 \phi^*(v_l)[a_l] \odot \mathcal{B}(u, \theta)\right)_{1 \leq l \leq L}.$$

The final procedure to compute the hypergradient, in the case that $w^{(Q)}(\theta)$ is obtained through algorithm (12), is detailed in Algorithm 3.

**Algorithm 3** Group Lasso Hypergradient (Reverse mode): **GLHypergradient**$(X, y, \theta, \lambda, C, Q)$

---

**Require:** Design matrix $X$, vector of outputs $y$, group structure $\theta$, number of inner iterations $Q$.

Initialize $u^{(0)}(\theta) \equiv 0 \in \mathbb{R}^{P \times L}$.

**for** $q = 1$ to $Q$ **do**

$\quad w^{(q-1)}(\theta) = (X^\top X + \epsilon \mathrm{Id}_p)^{-1}\big(X^\top y - \sum_{l=1}^{L} \theta_l \odot u_l^{(q-1)}(\theta)\big).$

$\quad v^{(q-1)}(\theta) = \big(\nabla \phi(u_l^{(q-1)}(\theta)) + \gamma \theta_l \odot w^{(q-1)}(\theta)\big)_{1 \le l \le L},$

$\quad u^{(q)}(\theta) = \big(\nabla \phi^*(v_l^{(q-1)}(\theta))\big)_{1 \le l \le L}.$

**end for**

$w^{(Q)}(\theta) = (X^\top X + \epsilon \mathrm{Id}_p)^{-1}\big(X^\top y - \sum_{l=1}^{L} \theta_l \odot u_l^{(Q)}(\theta)\big).$

**output** 1. $u^{(0)}(\theta), \ldots, u^{(Q)}(\theta), v^{(0)}(\theta), \ldots, v^{(Q)}(\theta), w^{(0)}(\theta), \ldots, w^{(Q)}(\theta).$

Initialize $z^{(Q)}(\theta) = (X^\top X + \epsilon \mathrm{Id}_p)^{-1}\nabla C(w^{(Q)}(\theta)),$

$\quad\quad\quad a_Q = -\big(\theta_l \odot z^{(Q)}(\theta)\big)_{1 \le l \le L},$

$\quad\quad\quad b_Q = -\big(u_l^{(Q)}(\theta) \odot z^{(Q)}(\theta)\big)_{1 \le l \le L}.$

**for** $q = Q - 1$ to $0$ **do**

$\quad w^{(q)} = \big(\nabla^2 \phi^*(v_l^{(q)}(\theta))[a_l^{(q+1)}]\big)_{1 \le l \le L},$

$\quad z^{(q)}(\theta) = (X^\top X + \epsilon \mathrm{Id}_p)^{-1} \sum_{l=1}^{L} \theta_l \odot w_l^{(q)},$

$\quad a^{(q)} = \big(\nabla^2 \phi(u_l^{(q)}(\theta))[w_l^{(q)}] - \gamma \theta_l \odot z^{(q)}(\theta)\big)_{1 \le l \le L},$

$\quad b^{(q)} = \gamma\big(-u_l^{(q)}(\theta) \odot z^{(q)}(\theta) + w_l^{(q)} \odot w^{(Q)}(\theta)\big)_{1 \le l \le L} + b^{(q+1)}.$

**end for**

**output** 2. Hypergradient $\nabla \mathcal{U}^{(Q)}(\theta) = b^{(0)}.$

---

**Algorithm 4** Bilevel learning of the Group Lasso structure through proxSAGA

---

**Require:** Vectors of outputs $\{y_t\}_{t=1}^{T}$, design matrices $\{X_t\}_{t=1}^{T}$, regularization parameter $\lambda > 0$, number of groups $L$, $C = \sum_{t=1}^{T} C_t / T$ defined in Problem (2.1) and $\Theta$ introduced in Problem 2.2.

Set the step-size $\gamma > 0$.

Initialize $\theta^{(0)}$.

Initialize $\tilde{G}_t = $ **GLHypergradient**$(X_t, y_t, \lambda, \theta^{(0)}, C_t, Q)$ for every $t \in \{1, \ldots, T\}$.

Initialize $d^{(0)} = (1/T) \sum_{t=1}^{T} \tilde{G}_t$.

**for** $k = 0$ to $K - 1$ **do**

$\quad$ Uniformly pick $t_k \in \{1, \ldots, T\}$

$\quad G_{t_k} = $ **GLHypergradient**$(X_{t_k}, y_{t_k}, \lambda, \theta^{(k)}, C_{t_k}, Q)$

$\quad \alpha^{(k)} = G_{t_k} - \tilde{G}_{t_k} + d^{(k)}$

$\quad d^{(k+1)} = (1/T)(G_{t_k} - \tilde{G}_{t_k}) + d^{(k)}$

$\quad \theta^{(k+1)} = \mathcal{P}_\Theta\big(\theta^{(k)} - \gamma \alpha^{(k)}\big)$

$\quad \tilde{G}_{t_k} = G_{t_k}$

**end for**

**output** Group-structure $\theta^{\mathrm{BiGL}} := \theta^{(K)}.$

---

## B.2 Overall Bilevel Scheme

The proposed bilevel scheme for learning the group structure is reported in Algorithm 4.

**Remark B.1.** The proposed scheme, which relies on the proxSAGA algorithm, requires the computation of the full gradient only once at the initialization step of $d^{(0)}$. In order to avoid its costly computation, we can initialize $\theta^{(0)}$ close to a saddle-point, such that $\theta^{(0)} = \mathcal{P}_\Theta\left((1/L)\mathbb{1}_{P \times L} + n\right)$ where $n$ is a small Gaussian perturbation. Hence, we can resort to the following approximate initialization: $d^{(0)} = 0_{P \times L}$ and $\tilde{G}_t = 0_{P \times L}$ for every $t \in \{1, \ldots, T\}$.

# C   Additional Results on Synthetic Data

Figure 5: The present results aim at complementing the ones displayed in Figure 1. The true features w*, consisting of 500 tasks, are displayed in the left plot and shown to exhibit 10 groups. The comparison of the validation and test error are reported in the middle and right figure respectively.