[Reviews · NeurIPS 2018]

Reviewer 1



The paper addresses the group lasso problem where the groups have to be learned (rather than being pre-supplied). This problem is cast as a (continuous) bilevel optimisation problem and the optimisation problem addressed by an algorithm devised by the authors. Experiments on synthetic data are given and support the usefulness of the presented method. Groups are discrete features so the authors have to relax the problem by removing the integrality constraint on group membership indicator variables. We are not given an idea of how much is "lost" by relaxing, but given the difficulty of the original problem I have no objection to solving a relaxation of the original problem. The paper is well-written (apart from many small errors - see below), the problem worth solving, the algorithm reasonable and the experimental results are positive. Using validation error to learn the groups is nice and straightforward. The empirical results given are also well chosen, particuarly the comparison with oracle GL. So I have difficulty in finding substantive criticisms of the paper. It's a typical NIPS submission: optimise to solve some machine learning problem. SMALL PROBLEMS 21: "a priori structure of the data". I found this confusing: what is a priori can not be found in the data. 25: we are only told later what \theta is so it is confusing here. 100: arbitrary -> arbitrarily 111: iterates -> iterations 135: the solve -> to solve 212: as less informative -> as uninformative 246,247: change to "... Q sufficiently permits reaching" 247: stress out -> stress 250: "Since a group ..." This sentence needs rewriting. Refs: Fix capitalisation: e.g. Lipschitz, Bregman AFTER DISCUSSION / FEEDBACK I remain positive about the paper!

Reviewer 2



The contributions of this paper consist in presenting a formulation for learning groups of variables in multitask linear prediction problems, and developing an optimization algorithm to solve the bilevel problem. The idea of learning group structure is not new. The authors mention other approaches in the literature, but do not compare with them. The formulation introduced by the authors is missing some discussion to make clear what is the meaning of the groups obtained in cases where there are few or only one task. Are all the zero variables put together in the same group? In addition, the group lasso penalty usually includes a factor on each group that is proportional to its size (see for example Meier et al. (2008) "The group lasso for logistic regression" JRSSSB). Missing this factor, it seems that the method might give preference to equal sized groups? The optimization algorithm and theoretical results are clearly written and sound. However, several optimization strategies have been used for solving problems like 3.1. The authors discuss some other approaches but I believe the motivation of why this new method is needed can be improved. Some of the details in the proof of Theorem 3.1 are referred to [2] but should be included for completeness. Numerical simulations and real data show the effectiveness of their method, but including other approaches for learning groups and prediction in multi-task learning can be included to make the contribution clearer. The real data example is brief. UPDATE: Thanks to the authors for their response. After reading the rebuttal letter, my questions have been clarified.

Reviewer 3



This is a strong paper that proposes a convincing solution to the open problem of selection of groups for group lasso. It proposes consistency guarantees (rates of convergence seem out of reach for such a difficult problem). The idea is to introduce a continuous relaxation of the mixed-integer problem, combined with recent results about bilevel optimization (Ochs et al.). Numerical experiments are convincing, even when the number of groups given as input overestimates the real number of groups. The paper opens a way to deal with a set of other losses and penalizations, which seems very promising. My only criticism is that the authors could do a comparison / parallel with relaxation techniques used for mixed-integer programming, since some heuristics used in this field might help or bring new ideas. I also wonder what would be the performance of a MIP solver directly applied to the problem, with the subproblem solved by an efficient convex solver, such a numerical comparison might motivate even further the approach of the paper.